# Minimum inhibitory concentrations of rifampin and isoniazid among multidrug and isoniazid resistant *Mycobacterium tuberculosis* in Ethiopia

Muluwork Getahun[1,2]*, Henry M. Blumberg[3], Gobena Ameni[4,5], Dereje Beyene[2], Russell R. Kempker[3]

1 TB and HIV Directorate, Ethiopian Public Health Institute, Addis Ababa, Ethiopia, 2 Department of Microbial, Cellular and Molecular Biology, Addis Ababa University, Addis Ababa, Ethiopia, 3 Division of Infectious Diseases, Department of Medicine, Emory University School of Medicine, Atlanta, Georgia, United States of America, 4 Aklilu Lemma Institute of Pathobiology, Addis Ababa University, Addis Ababa, Ethiopia, 5 Department of Veterinary Medicine, College of Agriculture and Veterinary Medicine, United Arab Emirates University, Al Ain, United Arab Emirates

* mimishaget@yahoo.com

**Data Availability Statement:** All relevant data files are available from the figshare database (https://figshare.com/articles/dataset/Minimum_inhibitory_

## Abstract

### Introduction

Traditionally, single critical concentrations of drugs are utilized for *Mycobacterium tuberculosis* (*Mtb*) drug susceptibility testing (DST); however, the level of drug resistance can impact treatment choices and outcomes. Mutations at the *katG* gene are the major genetic mutations in multidrug resistant (MDR) *Mtb* and usually associated with high level resistance. We assessed the minimum inhibitory concentrations (MICs) of MDR or rifampin resistant (RR) and isoniazid (INH) resistant *Mtb* isolates to determine the quantification of drug resistance among key anti-tuberculosis drugs.

### Methods

The study was conducted on stored *Mtb* isolates collected as part of a national drug resistance survey in Ethiopia. MIC values were determined using Sensititre™ MYCOTB plates. A line probe assay (MTBDR*plus*) was also performed to identify genetic determinants of resistance for all isolates.

### Results

MIC testing was performed on 74 *Mtb* isolates including 46 MDR, 2 RR and 26 INH phenotypically resistant isolates as determined by the Löwenstein Jensen (LJ) method. Four (15%) INH resistant *Mtb* isolates were detected as borderline rifampin resistance (MIC = 1 µg/ml) using MYCOTB MIC plates and no rifampin resistance mutations were detected by LPA. Among the 48 MDR/RR TB cases, 9 (19%) were rifabutin susceptible (MIC was between ≤0.25 and 0.5µg/ml). Additionally, the MIC for isoniazid was between 2–4 µg/ml

concentration_of_rifampin_rifabutin_and_
isoniazid/19749964); DOI: 10.6084/m9.figshare.
19749964.

**Funding:** This work was supported in part by the
Ethiopian Public Health Institute (EPHI) and the
National Institutes of Health (NIH) Fogarty
International Center (D43 TW009127). The funders
had no role in study design, data collection and
analysis, decision to publish, or preparation of the
manuscript.

**Competing interests:** The authors have declared
that no competing interests exist.

(moderate resistance) for 58% of MDR TB isolates and 95.6% (n = 25) of the isolates had
mutations at the *katG* gene.

## Conclusion

Our findings suggest a role for rifabutin treatment in a subset of RR TB patients, thus poten-
tially preserving an important drug class. The high proportion of moderate level INH resistant
among MDR *Mtb* isolates indicates the potential benefit of high dose isoniazid treatment in a
high proportion of *katG* gene harboring MDR *Mtb* isolates.

## Introduction

The effective management of drug resistant tuberculosis (TB) relies on the accurate detection
of resistance to guide appropriate treatment [1]. Drug susceptibility testing (DST) is an impor-
tant step in the design of effective treatment regimens as it guides clinical management of indi-
vidual patients and is essential for TB control program planning [2]. Traditionally, a single
critical concentration (CC) per drug has been utilized to determine if *Mycobacterium tubercu-
losis* (*Mtb*) isolates are resistant or susceptible to an individual drug; however, this approach
limits the ability to detect borderline resistance and can result in resistance misclassification.
Minimum inhibitory concentration (MIC) testing provides quantitative results that may help
guide the therapeutic decision-making process [3, 4]. High level drug resistance indicates a
drug is unlikely to have any clinical benefit; on the other hand, low-level drug resistance indi-
cates a possible benefit of a drug either with standard or increased dosing from a different
drug with the same class [3, 5].

The Sensititre™ MYCOTB plate (Trek Diagnostic Systems, Thermo Fisher Scientific, USA)
was developed to determine the MICs of key first- and second-line TB drugs that includes
ofloxacin, amikacin, moxifloxacin, linezolid, cycloserine, clofazimine, kanamycin, levofloxa-
cin, rifampin, capreomycin, para-aminosalicylicacsid, isoniazid, tedizolid, and rifabutin. Previ-
ous reports have found the Sensititre™ MYCOTB to be accurate method for performing DST
for both first- and second-line anti-TB drugs and result would be available within 14 days [6–
8]. The results allow the clinician to adjust dose or choose among the best available alternatives
to optimize therapy [3, 8, 9].

Certain mutations are more likely to confer high levels resistance whereas some may be
found in low level resistance. Mutations at the *inhA* promoter are mostly conferring low-level
resistance, while mutations in *katG* gene particularly at S315T are usually associated with high
level resistance [10, 11]. The *katG* mutations are the major genetic mutations in MDR TB [12]
and it had a wide range of MIC value that suggest some MDR patient might potentially
benefited from high dose treatment [13].

Rifabutin is an alternative first line anti-TB drug for TB-HIV co-infected patients [14]. As
indicated in previous studies, rifabutin based therapy had very low relapse rate for person co-
infected with TB and HIV [15, 16]. Some RR TB are susceptible for rifabutin and those cases
are more likely to harbor certain mutation than resistant cases for both rifampin and rifabutin
[17–19]. The presence or absence of cross resistance of rifampin with rifabutin may assist in
guiding the treatment choice for the rifabutin based therapy [20].

To our knowledge, in Ethiopia there is no previous data on MIC of key first line anti-TB
drugs (isoniazid, rifampin and rifabutin) for *Mtb* isolates including multidrug or rifampin
resistant (MDR/RR) and isoniazid (INH) resistant isolates. To help fill this knowledge gap, we
conducted a study to determine MIC values of MDR/RR TB and INH resistance *Mtb* isolates

collected from population-based drug resistance survey. The overall goal of our study is to provide information about the proportion of TB cases with low, moderate, and high-level resistance to INH and the proportion of rifabutin susceptible among RR *Mtb* isolates that could potentially benefit in MIC tailored regimen choice.

## Materials and methods

We utilized stored *Mtb* isolates to profile the MIC values of MDR/RR TB and INH resistant *Mtb* isolates. The study population has been described in detail previously [21]. Briefly, we utilized stored MDR/RR and INH resistant *Mtb* isolates collected for a national drug resistance survey (DRS) conducted between November 2011 and June 2013. The DRS enrolled all TB suspects with sputum smear positive samples from 32 health facilities throughout Ethiopia. The 32 health facilities were purposely selected to include facilities in each region throughout the country and to provide a representative sample of Ethiopia. Phenotypic DST was performed on all *Mtb* isolates using the indirect proportion Lowenstein-Jensen (LJ) method for first-line drugs including INH (0.2 μg/ml) and rifampin (RIF) (40 μg/ml). Interpretation of results was based on the proportion of the growth on control and drug containing media [22, 23]. The DRS found 67 MDR, 5 RR and 70 INH resistant *Mtb* isolates. Among these isolates, we selected 74 *Mtb* isolates (46 MDR, 2 RR and 26 INH resistant TB) based on subculture growth characteristics i.e pure *Mtb* growth, *Mtb* growth with contamination, no growth (S1 Table). All available MDR/RR and previously treated INH resistant *Mtb* isolate that had sufficient growth were undergo with MIC testing [22].

### Line probe assay

All stored MDR/RR and INH *Mtb* isolates had LPA testing performed. The Genotype MTBDR*plus* (Hain Lifescience, Nehren, Germany, version 2) assay was performed to evaluate for genetic determinants of resistance to rifampin and isoniazid. Extraction, amplification, detection, and result interpretation were performed as per the manufacturer's instructions [24].

### Minimum inhibitory concentration

After thawing frozen *Mtb* isolates, 100μl of vortexed allocate was subcultured on to Middlebrook 7H11 agar medium. Sensititre™ MYCOTB MIC plates were utilized for MIC testing for all included *Mtb* isolates. The MYCOTB plate contains lyophilized antibiotics of three key first-line drugs including isoniazid (0.25–4 μg/ml), rifampin (0.25–8 μg/ml) and rifabutin (0.25–8 μg/ml). MIC testing was performed according to the manufacturer's recommendation. Briefly, colonies were scraped, and a 0.5 McFarland standard was prepared using a saline-Tween solution. After 15 minutes, 100 microliters were transferred to 11ml of Middlebrook 7H9 broth containing oleic acid albumin-dextrose-catalase and vortexed for 30s. Each well of the MIC plate was inoculated with 100 microliters of the reconstituted sample aliquot. Plates were then covered with plastic seals and incubated at 37˚C, and visually monitored for growth at days 1, 2, 7, 10, 14 and 21. The MIC result was recorded on days 10 and subsequently on 21 if growth was not sufficient for interpretation. Blood agar and Middlebrook 7H10 plates were also inoculated with suspension to check contamination and read after 24 and 48 hours.

We classified *Mtb* isolates as either susceptible or resistant based on reported CCs [6, 7, 9, 25]. The lowest concentration with no visible growth was viewed using a magnified mirror and recorded as the MIC value. *Mtb* isolates were defined as resistant if the MIC was greater than the CC or susceptible if the MIC was less than or equal to the CC. For INH, we categorized into the following 3 groups based on MIC values: low level resistance (≤0.25μg/ml to

0.5μg/ml), moderate resistance (1μg/ml≥ MIC ≤4μg/ml) and high level resistance (MIC>4μg/ml). The upper cutoff values of moderate level resistance were defined based on a clinical trial report in which patients with such levels of resistance were having a comparable early bactericidal activity as sensitive strains [26].

### Ethical approval

We use a stored isolates. The study obtained ethical approval from the Addis Ababa University Ethics Committee.

### Statistical analysis

Descriptive statistics were used to describe the MIC values of MDR, RR and INH resistant *Mtb* isolates.

## Results

### Study isolates and patients

A total of 74 *Mtb* isolates from unique patients including 46 MDR, 2 RR and 26 INH phenotypically resistant isolates had MIC testing using MYCOTB Sensititre™ plates performed. Among the 74 unique individuals from which the samples came from, 58% were male and the mean age was 30 years. Forty six percent of patients (n = 34) had a history of prior anti-TB treatment with first-line drugs (2RHZE/4RH). Ninety five percent (n = 70) of the patients had human immunodeficiency virus (HIV) testing performed and 33% (n = 23) were HIV positive. Over 50% (n = 13) of the HIV positive individual had a history of prior anti-TB treatment (2RHZE/4RH).

### MIC testing

Among the 46 MDR, 2 RR and 26 INH resistant *Mtb* isolates, MYCOTB Sensititre™ MIC plate confirmed resistance in 46, 2 and 24 cases respectively (Table 1). Two INH resistance *Mtb* isolates as defined by LJ method were detected as susceptible using Sensititre™ MYCOTB MIC plate (MIC≤0.25 μg/ml) and mutations confirming INH resistance were detected using LPA. Four (15%) INH resistant *Mtb* isolates as defined by LJ method were borderline rifampin resistant TB at CC = 0.5 μg/ml using MYCOTB MIC plates and no rifampin resistance mutations were detected by LPA. The *Mtb* isolates with borderline rifampin resistant were recovered from patients with newly diagnosed TB.

### Rifabutin MIC distribution

The MIC of *Mtb* isolates to rifabutin and the corresponding mutation profiles are summarized in Table 2. Sixty two percent of the mutations occurred at codon 530–533 and 93% of rifabutin resistant *Mtb* isolate had MIC ≥ 4μg/ml. Among 48 MDR/RR *Mtb* isolates, 9 isolates (19%) were found to be rifabutin-susceptible (MIC≤0.5μg/ml), including 6 patients with a prior history of TB treatment and 4 from persons living with HIV. Six (66%) of the 9 rifampin-resistant and rifabutin-susceptible *Mtb* isolates were classified as rifampin resistance due to the absence of wild-type probes only. Furthermore, 33% of the rifampin-resistant and rifabutin-susceptible *Mtb* isolates had mutations detected by the MDRTB*plus* probe in the 513–519 codon regions.

**Table 1. Minimum inhibitory concentration of rifampin, rifabutin and isoniazid among multidrug resistant (MDR), rifampin resistant (RR) and isoniazid (INH) resistant _Mycobacterium tuberculosis_ isolates.**

| Drugs | DST result | MIC in µg/ml | Total N | Total % | Newly diagnosed, N | Retreatment, N |
|---|---|---|---|---|---|---|
| Rifampin | MDR | >8 | 46 | 100 | 21 | 27 |
| | RR TB | >8 | 2 | 100 | 1 | 1 |
| | INH resistant | 0.5 | 4 | 15 | 4 | 0 |
| | | ≤0.25 | 22 | 85 | 15 | 7 |
| Rifabutin | MDR | >8 | 18 | 39 | 4 | 14 |
| | | 8 | 7 | 15 | 3 | 4 |
| | | 4 | 9 | 20 | 9 | 0 |
| | | 2 | 2 | 4 | 1 | 1 |
| | | 1 | 2 | 4 | 1 | 1 |
| | | 0.5 | 1 | 2 | 0 | 1 |
| | | ≤0.25 | 7 | 15 | 2 | 5 |
| | RR TB | 2 | 1 | 50 | 0 | 1 |
| | | ≤0.25 | 1 | 50 | 1 | 0 |
| | INH resistance | ≤0.25 | 26 | 100 | 19 | 7 |
| Isoniazid | MDR | >4 | 19 | 41 | 8 | 11 |
| | | 4 | 25 | 54 | 10 | 15 |
| | | 2 | 2 | 4 | 2 | 0 |
| | INH resistance | >4 | 9 | 35 | 1 | 8 |
| | | 4 | 9 | 35 | 5 | 4 |
| | | 2 | 5 | 8 | 4 | 1 |
| | | 0.5 | 1 | 4 | 1 | 0 |
| | | ≤0.25 | 2 | 8 | 1 | 1 |

DST: drug susceptibility testing

**Table 2. Line probe assay result and minimum inhibitory concentration of rifabutin among multidrug resistant (MDR) and rifampin resistant (RR) _Mycobacterium tuberculosis_ isolates.**

| DST result | Wild type absent and mutation developed on MTBDR*plus* probe | MIC in µg/ml | | | | | | |
|---|---|---|---|---|---|---|---|---|
| | | ≤0.25 | 0.5 | 1 | 2 | 4 | 8 | >8 |
| MDR | Susceptible | 0 | 0 | 0 | 1 | 0 | 0 | 1 |
| | W3W4 | 2 | 0 | 0 | 0 | 1 | 0 | 0 |
| | W3W4MUT1 | 0 | 1 | 0 | 0 | 0 | 0 | 0 |
| | W4W7MUT1MUT2B | 0 | 0 | 1 | 0 | 0 | 0 | 0 |
| | W7 | 4 | 0 | 0 | 0 | 0 | 0 | 1 |
| | W7MUT2A | 1 | 0 | 0 | 0 | 0 | 0 | 1 |
| | W7MUT2B | 0 | 0 | 0 | 0 | 0 | 0 | 2 |
| | W8 | 0 | 0 | 0 | 0 | 1 | 0 | 1 |
| | W8MUT3 | 0 | 0 | 1 | 1 | 6 | 7 | 12 |
| | MUT3 | 0 | 0 | 0 | 0 | 1 | 0 | 0 |
| RR | Susceptible | 0 | 0 | 0 | 1 | 0 | 0 | 0 |
| | W7 | 1 | 0 | 0 | 0 | 0 | 0 | 0 |

DST: drug susceptibility testing

**Table 3. Frequency of isoniazid resistance *Mycobacterium tuberculosis* isolates with minimum inhibitory concentration and line probe assay result.**

| DST result | MIC in μg/ml | Total | Wild type absent and mutation developed on MTBDR*plus* probe | | | | |
|---|---|---|---|---|---|---|---|
| | | | Susceptible | inhAW1MUT1 | katGW1 | katGW1MUT1 | katGMUT1 |
| INH resistant | ≤0.25 | 2 | 0 | 1 | 0 | 0 | 1 |
| | 0.5 | 1 | 0 | 1 | 0 | 0 | 0 |
| | 2 | 5 | 2 | 0 | 0 | 3 | 0 |
| | 4 | 9 | 0 | 0 | 0 | 9 | 0 |
| | >4 | 9 | 0 | 0 | 0 | 8 | 1 |
| MDR | 2 | 2 | 0 | 0 | 0 | 2 | 0 |
| | 4 | 25 | 1 | 1 | 2 | 21 | 0 |
| | >4 | 19 | 1 | 0 | 0 | 17 | 1 |

DST: drug susceptibility testing

### Low and moderate level isoniazid *Mtb*

Among 46 MDR *Mtb* isolates, 58% (n = 27) had moderate level (MIC = 2–4 μg/ml) isoniazid resistance. Most (n = 25, 95.6%) moderate level isoniazid resistant *Mtb* isolates had mutations in the *katG* gene; of the remaining two isolates, one had mutations at the *inhA* gene mutations and the other isolates had no mutations detected by the LPA (Table 3). Among 26 INH resistant *Mtb* isolates, 3 (12%) had low level resistance (MIC ≤0.5μg/ml) including 2 with mutations detected by the MTBDR*plus* in the *inhA* gene.

## Discussion

Utilizing a national representative sample, we carried out the first study evaluating MIC testing of *Mtb* isolates in Ethiopia with results highlighting potential roles for rifabutin and high dose INH treatment in many patients with MDR TB. Our study reports borderline rifampin resistance among INH resistant *Mtb*, low level rifabutin resistance among RR *Mtb* and moderate level isoniazid resistance among *katG* gene harboring MDR *Mtb* isolates. Data on rifabutin susceptible RR *Mtb* would assist in guiding the treatment choice for the rifabutin-based therapy especially in HIV co-infected TB patients [15, 20]. Additionally, detections of borderline rifampin resistance using Sensititre™ MYCOTB MIC plate which mostly missed using phenotypic methods are vital to interrupt transmission [27–29]. Besides, information on the level of INH resistance is important to guide the treatment choice with high dose INH for MDR *Mtb* with *katG* mutations [26].

Rifabutin is considered as an alternative treatment option in place of rifampin in some persons with HIV and TB due to fewer drug-drug interactions with certain antiretroviral drugs [14]. In our study, 19% (n = 9) of the *Mtb* isolates that were MDR/RR were susceptible to rifabutin and 44% (n = 4) of them were person living with HIV; a finding that is in line with previous studies [17–19]. Studies show that the treatment outcomes and the rate of acquired resistance of rifabutin susceptible MDR TB patients can be improved by rifabutin-based therapy [16, 30]. A clinical trial on MDR TB patients who had been treated using individualized regimens reports a significantly higher treatment success rate for rifabutin based therapy (85.7%) than control group (52.4%) [20]. Another clinical trial indicates that TB-HIV co-infected patient who takes rifabutin twice a week had low relapse rate (1 of 20) than rifampin-based therapy (8 of 9) [15]. Additionally, mutations at codon 514, 516 and 522 are usually associated with rifabutin susceptibility even though the mutations alone had not always classified such cases [31]. In our study, we found that 33% rifampin resistant but

rifabutin susceptible was detected at codon 513–519; this is in line with previous studies [17–19]. Overall, our finding showed that the MIC of rifabutin susceptible MDR/RR *Mtb* isolates were between ≤0.25 and 0.5µg/ml which indicates the value of MIC determination in detecting those cases than the traditional DST which uses a single CC as well as frontline genotypic methods [31].

Borderline rifampin resistance is mostly associated with low-level rifampin resistance and can be missed by growth-based methods [27–29]. Our finding showed that 15% INH resistant *Mtb* isolates as defined by LJ method were borderline rifampin resistance at CC = 0.5 µg/ml (MIC = 1 µg/ml) using Sensititre™ MYCOTB MIC plate. We did not detect rifampin mutations using LPA suggesting the mutation might occur outside of rifampicin resistance-determining region. The WHO review indicates that a couple of rifampin borderline resistance was detected as susceptible using LJ method at the current CC (40 mg/L). Compare to LJ method, 36% of mutations at codon I572F and 28% of mutation at the six rifampicin resistance-determining region were detected as susceptible using LJ method [31]. The treatment outcome of patients with borderline rifampin resistance has been associated with treatment failures [32] and WHO recommends a MDR-TB regimen for their treatment [31]. Our findings indicate the importance of MIC in detecting borderline rifampin resistance which is mostly difficult to detect using phenotypic and frontline genotypic methods [27–29, 31].

World Health Organization recommended the inclusion of high dose isoniazid in MDR TB regimen to increase the treatment success [33]. Although the clinical efficacy of high dose isoniazid is not fully understood [34, 35]. In our study, the MIC of isoniazid was moderate level (1µg/ml≥ MIC ≤4µg/ml) for just over half of all MDR TB isolates, which is consistent with other studies [36, 37]. We found that 95.6% of moderate level isoniazid resistant TB had mutations in the *katG* gene and only one isolate had mutations at the *inhA* gene. A previous study reports MDR TB patients with *inhA* mutation have MIC value ranges from 0.05–4µg/ml and that they had comparable early bactericidal activity as sensitive strains. The high dose treatment(10–15 mg/kg daily) procedure a similar early bactericidal activity for participant with *inhA* resistance TB compared to isoniazid sensitive TB participant who were treated with standard dose (5mg/kg daily) [26]. As noted, the treatment outcome of low to moderate level isoniazid resistance with high dose isoniazid treatment would be increased if the treatment is guided by MIC results [38, 39]. Our finding suggests a high proportion of *katG* harboring MDR TB patients could benefit from high dose INH if MIC testing is performed.

Limitations of our study include the following: First, the MIC was performed in selected isolates especially for INH resistance. Additionally, treatment data were not available precluding us from evaluating outcomes based on certain phenotypic and genetic resistance. Therefore, a large scale study that includes treatment outcome and adverse events would provide additional needed data to inform evidence-based guidelines on the use of rifabutin and high dose isoniazid in the treatment of MDR-TB.

## Conclusions

We profile the MIC of rifampin, rifabutin and isoniazid using MYCOTB Sensititre™ MIC plate. Our findings suggest a potential role for rifabutin treatment in certain patients with RR TB and further studies are needed to explore the utility of providing a rifabutin for RR TB patients. Moreover, over half of MDR TB isolates had moderate level isoniazid resistant indicates the potential benefit of high dose isoniazid treatment in a high proportion of *katG* gene harboring MDR *Mtb* isolates if aligned with MIC values. Furthermore, additional studies on MIC testing are important to guiding management and allowing for personalized treatment in low-income countries like Ethiopia.

## Supporting information

**S1 Table. Subculture growth characteristics of multidrug resistant (MDR), rifampin resistant (RR) and isoniazid (INH) resistant *Mycobacterium tuberculosis* isolates in newly diagnosed and previously treated TB patients.** No growth indicates *Mtb* did not grow; not retrieved indicates the *Mtb* isolate could not be located and therefore, subculture was not performed; insufficient growth indicates that the colony count was between 3 and 10. (DOCX)

## Author Contributions

**Conceptualization:** Muluwork Getahun.

**Formal analysis:** Muluwork Getahun.

**Funding acquisition:** Henry M. Blumberg, Russell R. Kempker.

**Supervision:** Henry M. Blumberg, Gobena Ameni, Dereje Beyene, Russell R. Kempker.

**Writing – original draft:** Muluwork Getahun.

**Writing – review & editing:** Henry M. Blumberg, Gobena Ameni, Dereje Beyene, Russell R. Kempker.

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
