## [Decision Letter · Decision Letter 0]

27 Apr 2022

PONE-D-22-08463Minimum inhibitory concentrations of rifampin and isoniazid among multidrug and isoniazid resistant Mycobacterium tuberculosis isolatesPLOS ONE

Dear Dr.  Getahun,

Thank you for submitting your manuscript to PLOS ONE. After careful consideration, we feel that it has merit but does not fully meet PLOS ONE’s publication criteria as it currently stands. Therefore, we invite you to submit a revised version of the manuscript that addresses the points raised during the review process.Please address the comments from the reviewers and revise the title to include the geographical location. Please ensure that your decision is justified on PLOS ONE’s publication criteria and not, for example, on novelty or perceived impact.

We look forward to receiving your revised manuscript.

Kind regards,

Pradeep Kumar, Ph.D.

Academic Editor

PLOS ONE

Journal Requirements:

a) Did participants provide their written or verbal informed consent to participate in this study?

5, Please review your reference list to ensure that it is complete and correct. If you have cited papers that have been retracted, please include the rationale for doing so in the manuscript text, or remove these references and replace them with relevant current references. Any changes to the reference list should be mentioned in the rebuttal letter that accompanies your revised manuscript. If you need to cite a retracted article, indicate the article’s retracted status in the References list and also include a citation and full reference for the retraction notice.

Reviewers' comments:

Reviewer's Responses to Questions

**Comments to the Author**

1. Is the manuscript technically sound, and do the data support the conclusions?

Reviewer #1: Yes

Reviewer #2: Yes

Reviewer #3: Yes

2. Has the statistical analysis been performed appropriately and rigorously? 

Reviewer #1: I Don't Know

Reviewer #2: Yes

Reviewer #3: Yes

3. Have the authors made all data underlying the findings in their manuscript fully available?

Reviewer #1: Yes

Reviewer #2: Yes

Reviewer #3: Yes

4. Is the manuscript presented in an intelligible fashion and written in standard English?

Reviewer #1: Yes

Reviewer #2: Yes

Reviewer #3: Yes

5. Review Comments to the Author

Reviewer #1: Dear Author,

This article on "Minimum inhibitory concentrations of rifampin and isoniazid among multidrug and isoniazid resistant Mycobacterium tuberculosis isolates" was very well written, concise and easy to follow. In the context of Ethiopia, this type of study is highly significant. Since there was significant number of patients were identified with HIV positive, it was not discussed about the possible outcome/impact of HIV on MDR TB. However, please review the attached file for detailed comments.

Thanks

Reviewer #2: In the manuscript entitled “Minimum inhibitory concentrations of rifampin and isoniazid among multidrug and isoniazid resistant Mycobacterium tuberculosis isolates”, Getahun et al describe the drug sensitivity profiling of 74 clinical isolates against two first-line anti-TB drugs and rifabutin from different locations of Ethiopia. Using both growth inhibition and line probe assay, they estimated the minimum inhibitory concentration of the isolates and categorized them as drug sensitive or resistant. Importantly, the authors find that a fraction of the MDR/RR-TB remain susceptible to rifabutin indicating the importance of rifabutin in MDR-TB regimen. In addition, more than 50% of the MDR isolates are only moderately resistant to INH leading them to suggest a high dose INH treatment for MDR-TB.

The manuscript is brief and well written with sufficient details. However, I would still make a few points on the conclusion made by the authors.

Major comments:

1) Study sample size: The manuscript provides drug susceptibility data for 74 samples isolated from 32 health facilities from different locations of Ethiopia. However, the given sample size is far too small for making a representation of the TB drug susceptibility profile of Ethiopia. The authors could consider increasing the sample size.

2) Based on their observation, the authors recommend considering the use of rifabutin and high dose INH for the treatment of MDR-TB. It seems plausible. However, despite the available references on the use of rifabutin and high dose INH for treatment of MDR-TB, I feel that the sample size of the current study and the fraction (9 out of 48 MDR isolates, 19%) is yet very small to conclude this point.

3) The title of the manuscript looks very general. I would suggest revising the manuscript title to include the specific geographical location that is the country name.

Minor comments:

1) Add the full form of RFB in the first place

2) L94 : Underwent

3) L100: Subcultured

Reviewer #3: The manuscript entitled “Minimum inhibitory concentrations of rifampicin and isoniazid among multidrug and isoniazid resistant Mycobacterium tuberculosis isolates” has been reviewed and I would like to comment on the overall hypothesis and its execution. Authors here had tried to demonstrate that how important is the dose spectrum of the current anti-TB drugs. And how this can affect the treatment choices and outcome. This is a very serious concern where it is difficult to deicide on clinical breakpoints of these drugs for effective treatment outcome. My comments are below which would help in better understanding of the current study and if fulfilled, I would recommend acceptance.

1) My major concern is the use of LPA which; by now have several modifications after 2011 for determining the resistance of drugs. However, authors use a latest version of LPA which was modified in 2017/2018 but I am not sure if there are additional guidelines by Hains in 2019 to calculate the resistance of these drugs? Author should clarify this and give a detail about it.

2) What is the concentration used for sub- culturing these strains on 7H11 plates?

3) What is the difference between no growth on subculture and not retrieved group?

4) I am curious to know if there are inoculum differences when the strains were scrapped off from the 7H11 plate to resuspend for McFarland standard? For example, how much cfu/ml would it be in the borderline strains?

5) The detection rate of different type of mutations, for example mutations at different SNP’s have authors considered for detection rate, when determining the type of mutation?

6) I understand the authors mentioned about the data regarding treatment is not accessible but important to know that the mutation outside Rifampicin determining region in those two borderline mutant strains is a result of treatment failure/lost follow up.

7) Is there any possibility of predicting that mutations will occur in which region and whether they are correlated with the MIC? Are there any models? Authors may give description about it.

8) Are the suspects in the newly diagnosed category have a known TST and IGRA status?

9) I am always skeptical about using high concentrations of drugs, authors mention the use high concentrations of INH for treatment. Unfortunately, mechanistically it remains poorly understood. Also, it is important to note that INH is poorly penetrant and its possible to have side effects and emergence of persisters later. However, I do not deny that there are studies done but those TB treated subjects I am not sure if they followed up which remains a cause of concern.

6. PLOS authors have the option to publish the peer review history of their article (what does this mean?). If published, this will include your full peer review and any attached files.

Reviewer #1: No

Reviewer #2: **Yes: **Jees Sebastian

Reviewer #3: **Yes: **Vartika Sharma

---

## [Author Response · Author response to Decision Letter 0]

30 Jun 2022

Dear PLoS One, 

We appreciate the time and effort of the editor and reviewers in providing a detailed review of our manuscript. We have provided a point-by-point response below to outline how we have addressed the editor and reviewers’ comments. We thank you for the consideration of our work, 

Sincerely, 

Muluwork Getahun on behalf of the study team 

Response to Reviewers

Reviewer #1: 

1. Comment: This article on "Minimum inhibitory concentrations of rifampin and isoniazid among multidrug and isoniazid resistant Mycobacterium tuberculosis isolates" was very well written, concise and easy to follow. In the context of Ethiopia, this type of study is highly significant. Since there was significant number of patients were identified with HIV positive, it was not discussed about the possible outcome/impact of HIV on MDR TB. However, please review the attached file for detailed comments.

Response: Thank you for this comment. To highlight the topics of HIV and MDR-TB we have made revisions in the revised manuscript as follows. 

Page 9 line 187-189: “Among 48 MDR/RR Mtb isolates, 9 isolates (19%) were found to be rifabutin-susceptible (MIC≤0.5µg/ml), including 6 patients with a prior history of TB treatment and 4 from persons living with HIV.”

Page 10 line 214-215: “Rifabutin is considered as an alternative treatment option in place of rifampin in some persons with HIV and TB due to fewer drug-drug interactions with certain antiretroviral drugs.”

Page 10 line 226-227: “Another clinical trial indicates that TB-HIV co-infected patients who takes rifabutin twice a week had low relapse rate (1 of 20) compared to those on rifampin-based therapy (8 of 9) (Burman et al., 2006).” 

Reviewer #2: 

Major comments:

2. Comment: Study sample size: The manuscript provides drug susceptibility data for 74 samples isolated from 32 health facilities from different locations of Ethiopia. However, the given sample size is far too small for making a representation of the TB drug susceptibility profile of Ethiopia. The authors could consider increasing the sample size.

Response: We included all the available MDR/RR Mtb isolates, all previously treated isoniazid resistant Mtb and 40% of newly diagnosed isoniazid resistant Mtb isolates that were collected from the national drug resistance survey. The remaining 22 newly diagnosed isoniazid Mtb isolates did not undergo testing with MIC due to the shortage of MIC plates

3. Comment: Based on their observation, the authors recommend considering the use of rifabutin and high dose INH for the treatment of MDR-TB. It seems plausible. However, despite the available references on the use of rifabutin and high dose INH for treatment of MDR-TB, I feel that the sample size of the current study and the fraction (9 out of 48 MDR isolates, 19%) is yet very small to conclude this point.

Response: We agree with the reviewer’s comment. While our study highlights the potential role for rifabutin for MDR TB patients the small sample size precludes broad generalizability. Based on the reviewer’s comment and to make this point clear, we have added in the following sentence to the limitations paragraph in the revised manuscript. 

Page 12 line 264-267: “Therefore, a large scale study that includes treatment outcome and adverse events would provide additional needed data to inform evidence-based guidelines on the use of rifabutin and high dose isoniazid in the treatment of MDR-TB. “

4. Comment: The title of the manuscript looks very general. I would suggest revising the manuscript title to include the specific geographical location that is the country name.

Response: Based on the reviewer’s comment, we have revised the title of the manuscript to include the location of the work. The new title of the manuscript (line 1, page 1) is “Minimum inhibitory concentrations of rifampin and isoniazid among multidrug and isoniazid resistant Mycobacterium tuberculosis in Ethiopia”

Minor comments:

5. Comment. Add the full form of RFB in the first place

Response: Thank you for this comment. In response to this comment, we are now using “rifabutin” rather than RFB throughout the manuscript. 

6. Comment L94: Underwent

Response: We have revised this sentence in the revised manuscript. Page 6 line 130: “All stored MDR/RR and INH-resistant Mtb isolates had LPA testing performed.”

7. Comment L100: Subcultured

Response: We have revised this sentence in the revised manuscript. Page 6 line 136-137: “After thawing frozen Mtb isolates, 100 µl of vortexed allocate was subcultured on to Middlebrook 7H11 agar medium. 

Reviewer #3: 

8. Comment: My major concern is the use of LPA which; by now have several modifications after 2011 for determining the resistance of drugs. However, authors use a latest version of LPA which was modified in 2017/2018 but I am not sure if there are additional guidelines by Hains in 2019 to calculate the resistance of these drugs? Author should clarify this and give a detail about it.

Response: Our study utilized the latest version of the Hains LPA test, and there have been no further modifications since that time. 

9. Comment: What is the concentration used for sub-culturing these strains on 7H11 plates?

Response: We used 100 µl of vortexed Mtb isolates, and we have highlighted this in the manuscript as outlined below. 

Page 6 line 136-137: “After thawing frozen Mtb isolates, 100 µl of vortexed allocate was subcultured on to Middlebrook 7H11 agar medium.” 

10. Comment: What is the difference between no growth on subculture and not retrieved group?

Response: No growth indicates that the subculture was performed but Mtb did not grow. While, not retrieved indicates the isolates were not located and available and therefore, a subculture was not performed. We have made the following revision in the revised manuscript:

Page 26 line 499-501: “No growth indicates Mtb did not grow; not retrieved indicates the Mtb isolate could not be located and therefore, subculture was not performed; insufficient growth indicates that the colony count was between 3 and 10.” 

11. Comment: I am curious to know if there are inoculum differences when the strains were scrapped off from the 7H11 plate to resuspend for McFarland standard? For example, how much cfu/ml would it be in the borderline strains?

Response: We used a 0.5 McFarland standard for all isolates which equates to CFU/ml of 1.5 x 108. This ensured that all inoculum sizes were the same. 

12. Comment: The detection rate of different type of mutations, for example mutations at different SNP’s have authors considered for detection rate, when determining the type of mutation? 

Response: We have outlined the mutations identified by LPA in Table 2 and have revised the following sentences: 

Page 9 line 185-186: The MICs of Mtb isolates to rifabutin and the corresponding mutation profiles are summarized in Table 2. The identified mutation profiles are described under comment 14. 

13. Comment: I understand the authors mentioned about the data regarding treatment is not accessible but important to know that the mutation outside Rifampicin determining region in those two borderline mutant strains is a result of treatment failure/lost follow up.

Response: The borderline mutant strains were present among patients with newly diagnosed INH-resistant TB. The following revision was made based on the reviewer’s comment:

Page 8 line 182-183: The Mtb isolates with borderline rifampin resistant were recovered from patients with newly diagnosed TB.

14. Comment: Is there any possibility of predicting that mutations will occur in which region and whether they are correlated with the MIC? Are there any models? Authors may give description about it.

Response: Mutations at codon 530-533 likely had a higher MIC to rifabutin. The information is described in the revised manuscript as follows: 

Page 9 line 186-187: Sixty two percent of the mutations occurred at codon 530-533 and 93% of rifabutin resistant Mtb isolate had MIC ≥ 4µg/ml.

Page 9 line 190-192: Six (67%) of 9 rifampin-resistant and rifabutin-susceptible Mtb isolates were classified as rifampin resistance due to the absence of wild-type probes only.

15. Comment: Are the suspects in the newly diagnosed category have a known TST and IGRA status?

Response: Thank you for this comment. Unfortunately, we did not have this clinical information on TST and IGRA results. Additionally, it should be noted that the TST and IGRA are not routinely performed among persons with suspected TB in Ethiopia. 

16. Comment: I am always skeptical about using high concentrations of drugs, authors mention the use high concentrations of INH for treatment. Unfortunately, mechanistically it remains poorly understood. Also, it is important to note that INH is poorly penetrant and its possible to have side effects and emergence of persisters later. However, I do not deny that there are studies done but those TB treated subjects I am not sure if they followed up which remains a cause of concern.

Response: We appreciate the reviewer’s comments and agree with this concern. Regardless of MIC testing results, high dose isoniazid is often part of an empiric MDR TB treatment regimen in low income countries such as Ethiopia. As noted, the treatment outcome of low to moderate level isoniazid resistance with high dose isoniazid treatment would be increased if the treatment is guided by MIC results (Moulding, 1981, Lange et al., 2014). Our study highlights the potential role of the use of high dose INH treatment in many patients with MDR TB if an MIC test guided approach is implemented. 

Additionally, a large scale phase II clinical trial on “High-Dose Isoniazid Among Adult Patients With Different Genetic Genetic Variants of INH-Resistant Tuberculosis (TB)” would provide the needed evidence. (https://clinicaltrials.gov/ct2/show/NCT01936831?cond=isoniazid&draw=3&rank=24)

Based on the reviewer’s comment, the manuscript has been revised as follows: 

Page 12 line 264-267: Therefore, a large scale study that includes treatment outcome and adverse events would provide additional needed data to inform evidence-based guidelines on the use of rifabutin and high dose isoniazid in the treatment of MDR-TB.

---

## [Decision Letter · Decision Letter 1]

30 Aug 2022

Minimum inhibitory concentrations of rifampin and isoniazid among multidrug and isoniazid resistant Mycobacterium tuberculosis in Ethiopia.

PONE-D-22-08463R1

Dear Dr. Getahun,

We’re pleased to inform you that your manuscript has been judged scientifically suitable for publication and will be formally accepted for publication once it meets all outstanding technical requirements.

Kind regards,

Dwij Raj Bhatta, PhD

Academic Editor

PLOS ONE

Additional Editor Comments (optional):

Authors have included all the points in revised manuscript as suggested by reviewers! Such revision has been verified by original reviewers and suggested for acceptance of the current version, threfore this highly important research Article on MDR MTB isolates and rifamicin and isoniazid MIC determination , alternative treatment options for MDR TB be accepted for publication!

Reviewers' comments:

Reviewer's Responses to Questions

**Comments to the Author**

1. If the authors have adequately addressed your comments raised in a previous round of review and you feel that this manuscript is now acceptable for publication, you may indicate that here to bypass the “Comments to the Author” section, enter your conflict of interest statement in the “Confidential to Editor” section, and submit your "Accept" recommendation.

Reviewer #2: All comments have been addressed

Reviewer #3: All comments have been addressed

2. Is the manuscript technically sound, and do the data support the conclusions?

Reviewer #2: Yes

Reviewer #3: Yes

3. Has the statistical analysis been performed appropriately and rigorously? 

Reviewer #2: Yes

Reviewer #3: Yes

4. Have the authors made all data underlying the findings in their manuscript fully available?

Reviewer #2: Yes

Reviewer #3: Yes

5. Is the manuscript presented in an intelligible fashion and written in standard English?

Reviewer #2: Yes

Reviewer #3: No

6. Review Comments to the Author

Reviewer #2: (No Response)

Reviewer #3: (No Response)

7. PLOS authors have the option to publish the peer review history of their article (what does this mean?). If published, this will include your full peer review and any attached files.

Reviewer #2: **Yes: **Jees Sebastian

Reviewer #3: No

---

## [Editor Report · Acceptance letter]

4 Sep 2022

PONE-D-22-08463R1 

Minimum inhibitory concentrations of rifampin and isoniazid among multidrug and isoniazid resistant *Mycobacterium tuberculosis* in Ethiopia. 

Dear Dr. Getahun:

I'm pleased to inform you that your manuscript has been deemed suitable for publication in PLOS ONE. Congratulations! Your manuscript is now with our production department. 

Kind regards, 

on behalf of

Professor Dwij Raj Bhatta 

Academic Editor

PLOS ONE